# Using the Health Belief Model to Identify Predictors of COVID-19 Vaccine Acceptance among a Sample of Pregnant Women in the U.S.: A Cross-Sectional Survey

**DOI:** 10.3390/vaccines10060842

**Published:** 2022-05-25

**Authors:** Aubrey Jones, Dorothy Wallis

**Affiliations:** 1College of Social Work, The University of Kentucky, Lexington, KY 40506, USA; 2College of Social Work, The University of Tennessee, Knoxville, TN 37996, USA; sbr328@vols.utk.edu

**Keywords:** COVID-19, pregnancy, health belief model

## Abstract

The aim of the study was to identify factors that predict acceptance of the COVID-19 vaccine among pregnant women. Using the Health Belief Model, the authors administered a cross-sectional survey of pregnant and postpartum women in the United States during the COVID-19 pandemic. Overall, 227 women agreed to participate and completed the entire survey. Over half (59%) the participants had received the COVID-19 vaccine at the time of the study. Perceived barriers to vaccination (*p* < 0.001) and perceived benefits (*p* < 0.001) to vaccination were statistically significant predictors of vaccination. Trust in healthcare providers was also statistically predictive of vaccination (*p* = 0.001). Binary regression results were statistically significant (χ2(9) = 79.90, *p* < 0.001), suggesting that perceived benefits, barriers, severity, and susceptibility scores had a statistically significant effect on the odds of a participant being vaccinated. Results indicate a need for increased patient education regarding COVID-19 vaccination during pregnancy, including the benefits of vaccination for mother and fetus.

## 1. Introduction

Vaccination during pregnancy is not uncommon in the United States. Currently, the Center for Disease Control (CDC) recommends pregnant women receive the flu vaccine and Tdap vaccine while pregnant to protect the woman and her fetus from potential complications due to the flu and pertussis [1]. On 30 January 2020, a new viral threat emerged, and the WHO Emergency Committee declared a global health emergency due to the international rise of COVID-19 cases [2]. Less than two months later, on 11 March, the WHO declared the COVID-19 outbreak a pandemic [3]. As infections and death rates continued to climb, the development of a vaccine became a priority. In July and August 2020, respectively, Moderna and Pfizer published initial clinical trial data on their mRNA vaccines. Although pregnant women were excluded from clinical trials, the CDC, the American College of Obstetricians and Gynecologists, and the World Health Organization all recommend the COVID-19 vaccine for pregnant women [4,5,6]. Pregnant women are at an increased risk of severe illness and death from COVID-19 compared to non-pregnant people (6). Furthermore, developing COVID-19 while pregnant increases the risk of poor pregnancy outcomes, including preterm birth [6]. As the risks of COVID-19 are severe for pregnant women, the COVID-19 vaccine is recommended for women who are trying to become pregnant or are currently pregnant or breastfeeding. Despite the threat of COVID-19, hesitancy regarding receiving the vaccine has been widespread. Currently, only 31% of pregnant women have received the vaccine despite the known benefits of receiving the vaccine while pregnant [7]. Because COVID-19 is an ongoing threat to the health and safety of pregnant women and their offspring, the authors sought to better understand vaccine decision making as it pertains to pregnant women’s acceptance or rejection of the COVID-19 vaccine. We used the Health Belief Model (HBM) as our guiding theoretical framework for this study.

### Health Belief Model

Established in the early 1950s, the HBM has been used to understand the failure of people to adopt prevention strategies or screening tests for early detection of disease. Derived from psychological and behavioral theories, the HBM suggests that a person’s belief in a personal threat of illness or disease together with their belief in the effectiveness of a health behavior or action will predict the likelihood of a person adopting the behavior [8,9,10]. There are two foundational components of HBM: first, the desire to avoid illness, or get well if already experiencing illness, and second, the belief that a specific health action will prevent, or cure illness. According to the HBM, the course of action taken is influenced by the persons perception of the benefits and barriers related to the health behavior. There are six constructs included in the HBM: perceived susceptibility; perceived severity; perceived benefits; perceived barriers; cue to action; self-efficacy [11]. The first four were developed as the original tenants while the final two were added as HBM research evolved.

The HBM has been used to assess many differing health outcomes among pregnant women. Among pregnant women, HBM has been shown to be an effective framework regarding factors for improving nutrition and weight control and behaviors [12,13], promoting physical activity [14,15], preventing self-medication during pregnancy [16], and promoting self-care during pregnancy [17]. These decisions a woman makes during her pregnancy, including assessing the perceived benefits and barriers to adopting new health behaviors, can have an impact on her overall pregnancy health and outcomes. These can be key indicators for practitioners and researchers alike in promoting health outcomes for pregnant women.

The HBM has been used to evaluate health behaviors in various contexts, and the use of HBM with pregnant women’s acceptance of vaccines is not novel. Few studies have examined pregnant women’s acceptance of the influenza vaccine while pregnant [18,19] and others have used the HBM to assess acceptance of the Pertussis vaccine while pregnant [20]. In China, researchers applied the HBM to pregnant women’s COVID-19 vaccine intentions [21]. This research was conducted prior to vaccines being widely available and occurred while China engaged in greater public health measures such as continued isolation periods [22]. Researchers in this study found that vaccine acceptance was associated with being younger in age, living in the western region of China, having a lower level of education, being later in pregnancy, having a high knowledge of COVID-19’s impacts, and having a high levels of perceived susceptibility and benefits, and lower levels of perceived barriers [21]. While no other studies specifically use the HBM to assess pregnant women’s receipt of the COVID-19 vaccine, other international studies have found that women had various reasons for refusing the vaccination. Concerns about vaccine safety and general mistrust of vaccines, refusal of other recommended vaccines, and fear of side-effects were noted as top indicators for vaccine refusal [23,24,25,26].

The use of HBM with acceptance of the COVID-19 vaccine among pregnant women remains novel as the status of the pandemic changes and the recommendations for vaccination and prevention measures change nationally. The current study explores how different dimensions of the HBM can be used to predict vaccination receipt among pregnant women during a time when the COVID-19 vaccine was becoming more nationally available. Utilizing HBM in the context of vaccine decision making has the potential to inform interventions promoting health information and decisions among pregnant and postpartum women.

## 2. Materials and Methods

The University of South Dakota (A.J.’s previous institution) approved the research, IRB-21-106. Consent was obtained by having participants click yes to agree to participate and continuing to the survey. This was written consent by checking the “yes I agree” box on the survey. This study sought to examine demographic and behavioral factors in relation to COVID-19 vaccination during pregnancy. Once participants agreed by clicking “yes,” they were taken to the first page of the survey.

### 2.1. Design and Sample

This study is cross-sectional using a convenience sample of women who identified as pregnant or within six-months postpartum at the time of the survey. Other inclusion criteria included English speaking and being aged 18 years or older. Participants were recruited from mid-May to the end of June 2021 using the social media sites Facebook and Twitter. Targeted Facebook Ads were utilized to broaden the reach of potential participants. Additionally, single postings were made on Facebook group pages for pregnant women and new mothers. Participants in these identified groups self-selected into the study. The posts were shareable and could be spread by other Facebook users. Recruitment messaging along with a QR code were shared on Twitter by using hashtags and allowing for the post to be shared. Participants could enter to win 1 of 4 USD 25 giftcards for participating in the survey.

### 2.2. Survey Security

As online data collection becomes more common, bots have been known to invade online surveys, particularly when an incentive is involved. As such, it was important to the researchers to create systems that would detect bots from the outset. Using Qualtrics, the researchers were able to implement captcha. Captcha is a type of challenge-response that works to differentiate humans from robotic software programs and hacking codes [27]. Additionally, we used the ballot stuffing measure, which allows the research team to identify duplicate respondents. Furthermore, we utilized “Expert Review Fraud Detection” which is a component of Qualtrics that tracks patterns associated with bots. Responses are flagged as fraudulent, allowing the researcher to further examine the respondents identified as fraud and take one of the following steps: (1) Discard; (2) redirect response for analysis separately; (3) Flag and filter responses; (4) analyze the number of fraudulent responses and break them down by duplicates and bots [28]. In this study, the authors opted to discard any responses identified as a bot, thus not including those in the final analysis.

### 2.3. Measurement Tools

After informed consent was obtained, the self-administered questionnaire was provided. In addition to demographic questions, the questionnaire included a series of Likert items assessing HBM-related-constructs adapted from Fridman et al. [29]. Additionally, Henninger et al. developed a Health Belief Model Domain and Survey Item based on the HBM five factors predictive of health behavior change to assess influenza vaccination during pregnancy [30]. The author adapted the questionnaire to address COVID-19 vaccination during pregnancy. This model of the HBM and questions included in this study are consistent with other studies examining vaccination during pregnancy.

### 2.4. Outcome Variable: COVID-19 Vaccination Status

The outcome variable of interest for this study was whether or not the participant endorsed receipt of the COVID-19 vaccine. This variable was coded as 0 for “No” and 1 for “Yes.” The primary a priori hypotheses are as follows:

**H1.** 
*Perceived barriers will be associated with decreased vaccination endorsement;*


**H2.** 
*Perceived benefits will be positively associated with vaccination;*


**H3.** 
*Perceived severity will be positively associated with vaccination;*


**H4.** 
*Perceived susceptibility will be positively associated with vaccination.*


### 2.5. Questionnaire Addressing Health Belief Model Dimensions

Adapted from the HBM questionnaires predicting H1N1 and influenza vaccination during pregnancy, the questionnaire contained 13 Likert-scale items assessing perceived barriers, benefits, susceptibility, and severity of the COVID-19 virus and vaccines [20,27]. Four statements assessed perceived barriers to vaccination for the mother and her infant (e.g., I think the COVID-19 vaccine is harmful for my baby); five statements evaluated the perceived severity of the COVID-19 virus for the mother and her infant (e.g., COVID-19 is highly contagious); three statements assessed perceived susceptibility to the COVID-19 virus (e.g., I think pregnant women would get sicker from COVID-19); and finally, one question measured perceived benefits of the COVID-19 vaccine (e.g., I think the vaccine for COVID-19 protects me).

All items for each individual scale score were assessed using a Likert-scale (1 = Strongly Disagree to 5 = Strongly Agree). The individual questions in each category were totaled, and the scores ranged as follows: (1) Perceived barriers to the COVID-19 vaccine has a score ranging from 4 to 20; (2) Perceived benefits to the COVID-19 vaccine has a score ranging from 1 to 5; (3) Perceived susceptibility to the COVID-19 virus has a score ranging from 3 to 15; (4) Perceived severity of the COVID-19 virus has a score ranging from 5 to 25. Higher scores indicated higher levels of perceived susceptibility to the COVID-19 virus, higher perceived severity of the COVID-19 virus, higher perceived barriers to the COVID-19 vaccine, and higher perceived benefits of the COVID-19 vaccine. These individual sub-scale scores were used as the primary predictor variables in the model. The Cronbach’s alpha for the HBM model was 0.68, indicating adequate reliability in this sample [31]. All statements from the questionnaire are listed in Table 1.

### 2.6. Pregnancy, Medical, and Demographic Information

Demographic variables were collected to assess differences between participants who were and were not vaccinated. Whether or not the participant was currently pregnant or postpartum was measured and used as a control variable, with postpartum coded as the reference category (0) and pregnant coded as (1). Additional pregnancy variables were collected and used to assess group differences. These included the number of pregnancies a participant has had, measured as a categorical variable ranging from (1), this being their first pregnancy, to (6), having more than five pregnancies. Participants were also asked if they had experienced a past pregnancy loss and could answer “Yes,” “No,” or “Unsure.”

Participants were asked if their prenatal provider was an OBGYN, midwife, or other, and they were given the option to choose not to respond. They were also asked whether they trusted their provider to give them accurate medical information. The answers for this were coded as an ordinal variable ranging from “Strongly Agree” (coded as 6) to “Strongly Disagree” (coded as 0). Participants were also asked if they received all recommended vaccinations during pregnancy and answers were “No,” “Some,” “Most,” and “All.” Relating to other vaccines, participants were asked if they usually got the flu shot when not pregnant (No, Some of the Time, Most of the Time, Yes) or whether they had ever had the HPV vaccines (Yes, Maybe, No, Unsure). Finally, participants were asked two questions about whether they had personally had COVID-19 (“Yes,” “No,” “Not diagnosed but I think I had it”), or whether they personally knew someone who had contracted COVID-19 (Yes, No).

Final questions used for comparison between groups included participant demographic information. Age was collected and measured as a scale variable. Participants were also asked questions regarding their race (white, Black or African American, Asian, Native American or Native Alaskan, Pacific Islander, other) and their ethnicity (Hispanic or Latino and non-Hispanic or Latino). For the race and ethnicity categories, participants were given the option to “Select all that apply.” Additionally, participants were asked about their level of education, which was measured as a categorical variable ranging from some high school to a doctoral degree.

### 2.7. Sample Size Calculation

Between December of 2020 and June of 2021, there were over 130,000 women who were pregnant in the U.S. [32]. This serves as the larger population, as the research for consideration was vaccine decision making. Statistical power analysis for the study was conducted a priori using G*Power, and it was found that in logistic regression, to see a small effect with a benchmark of an odds ratio of 1.68, the total sample should at minimum be 104 [33]. The post hoc power analysis, with all 11 predictor variables and covariates included in the model, was determined to be 0.88.

### 2.8. Data Analysis Plan

Data were analyzed using IBM SPSS Statistics version 27.0.0.0. Descriptive statistics were generated on the sample, and Mann–Whitney U-Tests were conducted to assess any differences between participants who were and were not vaccinated. Binary logistic regression was run to assess whether the subscales of perceived susceptibility, perceived severity, perceived barriers, and perceived benefits contributed to a woman’s receipt of the COVID-19 vaccine, controlling for being pregnant vs. postpartum, age, race, and for significant covariates identified by the Mann–Whitney tests. For the regression model, McFadden’s R^2^ was used to examine model fit. Cohen’s *d* was used to report effect size to compare the probability as a function of either accepting the vaccine or not accepting the vaccine [33].

## 3. Results

### 3.1. Descriptive Statistics

Table 2 provides a descriptive look into the sample characteristics, separated by vaccination status. Overall, 227 women completed all of the Heath Belief Model questionnaire and were included in the final analysis. Among the participants, 134 (59.0%) were vaccinated against the COVID-19 virus. The majority of the sample was White (*n* = 163, 71.8%). The overall sample had a mean age of 29.61 (SD = 3.89). The majority of the sample (52.0%) had at least a bachelor’s degree.

Over 80% of participants were pregnant at the time of the survey, with the remaining being within six weeks postpartum. Over half reported that this was their first pregnancy. Of the participants, 63.9% reported they received all of the recommended vaccines during their current pregnancy. Similarly, most of the participants (60.4%) reported that they normally receive the flu vaccine when they are not pregnant, and 56.4% reported receiving the HPV vaccine at some point in their lifetime. When asked about their prenatal care provider, over 85% of women reported they saw an OBGYN. When asked “Do you trust your medical provider to give you accurate information that will benefit your health and your infant’s health?” 90.3% of participants reported that they “Somewhat Agreed” to “Strongly Agreed” with that statement. Participants were also asked if they had been diagnosed with COVID-19 during the pandemic. A large majority reported they had not. However, over 60% reported they personally knew someone who contracted COVID-19.

In calculating the Health Belief Model in relation to COVID-19, the average scale scores were as follows: Perceived Benefits of the COVID-19 Vaccine (M = 3.88, SD = 1.05, Min = 1.00, Max = 5.00); Perceived Barriers of the COVID-19 Vaccine (M = 11.18, SD = 3.50, Min = 4.00, Max = 20.00), Perceived Susceptibility to COVID-19 Virus (M = 10.95, SD = 2.33, Min = 3.00, Max = 15.00), and Perceived Severity of the COVID-19 Virus (M = 20.08, SD = 38.5, Min = 5.00, Max = 25.00). Higher scores indicated higher perceived benefits, barriers, susceptibility, and severity.

### 3.2. Vaccinated and Unvaccinated Participant Comparisons

Mann–Whitney U-Tests were conducted on the above descriptive statistics to assess whether any of the demographic variables varied by participants who were vaccinated versus those who were unvaccinated. Of all the demographic and health information assessed, education, knowing someone who contracted COVID-19, trust in the medical provider, and receiving all recommended pregnancy vaccines were statistically significantly different across groups (See Table 2). Vaccinated participants had higher levels of education (*p* < 0.001), higher trust in their medical care provider (*p* = 0.001), higher rates of receiving all recommended vaccinations during pregnancy (*p* < 0.001), and higher rates of knowing someone personally who had COVID-19 (*p* < 0.001) when compared to unvaccinated participants. No other demographic variables varied significantly across groups. The four subscales of the Health Belief Model varied significantly across groups. Those who were vaccinated perceived higher benefits (*p* < 0.001) and lower barriers (*p* < 0.001) to the COVID-19 vaccine, as well as higher rates of susceptibility (*p* = 0.049) and higher rates of severity (*p* = 0.003) of the COVID-19 virus.

### 3.3. Regression Results

A binary logistic regression was conducted to examine whether perceived benefits, barriers, severity, and susceptibility had a significant effect on the odds of a participant being vaccinated, when controlling for whether or not the participant was pregnant or postpartum, their education level, receiving all recommended vaccines while pregnant, knowing someone with COVID-19, their trust in their provider, age, and race. Control variables were chosen based upon significant demographic variables between vaccinated and not vaccinated participants, and while age and race were not statistically different between groups, they were added for additional levels of control. Multinomial variables were dichotomized based upon directionality significance from the Mann–Whitney tests. Education was dichotomized into 0 (some college, high school diploma/GED, and some high school) and 1 (bachelor’s, master’s, or doctoral degrees). Receiving all recommended vaccines during pregnancy was dichotomized into 0 (received no recommended vaccines, some recommended vaccines, and most recommended vaccines) and 1 (received all recommended vaccines). Trust in provider to give accurate medical information was dichotomized into 0 (strongly disagree to neither agree nor disagree) and 1 (somewhat agree to strongly agree). Finally, race was dichotomized into 0 (white participants) and 1 (participants from minoritized populations.) All values coded as 0 served as the reference category. Table 3 provides the regression results.

The assumption of absence of multicollinearity was examined, and Variance Inflation Factors (VIFs) were calculated to detect the presence of multicollinearity between predictors. High VIFs indicate increased effects of multicollinearity, with VIFs of greater than five being a cause for concern [34]. All predictors in the regression model have VIFs of less than five. The regression model was evaluated based on an alpha value of 0.05. The overall model was statistically significant (χ^2^ (11) = 88.58, *p* < 0.001), suggesting that perceived benefits, barriers, severity, and susceptibility scores had a statistically significant effect on the odds of a participant being vaccinated, when controlling for other variables within the model. McFadden’s R-squared was calculated to examine the model fit, where values of greater than 0.2 are indicative of models with excellent fit [35]. The McFadden R-squared value calculated for this model was 0.29.

Of the examined independent variables, only the perceived barriers of receiving the COVID-19 vaccine and perceived benefits of receiving the COVID-19 vaccine had a statistically significant result. The effect of perceived benefits was statistically significant (*B* = 0.66, OR = 1.93, *p* < 0.01), indicating that a one unit increase in perceived benefits increased the odds of being vaccinated by approximately 93%. The effect of perceived barriers was also statistically significant (*B* = −0.20, OR = 0.82, *p* = 0.01), indicating that a one-unit increase in perceived barriers of the COVID-19 vaccine decreased the odds of being vaccinated by 18%. Neither the perceived susceptibility to the COVID-19 virus (*B* = 0.03, OR = 1.03, *p* = 0.83) nor the perceived severity of the COVID-19 virus (*B* = 0.01, OR = 1.17, *p* = 0.79) had a significant effect on the participant being vaccinated for COVID-19. There were four significant covariates predicting whether or not a participant received the COVID-19 vaccination. Similar to the results found in the Mann–Whitney tests, participants who received all recommended vaccines during pregnancy were 365% more likely to also receive the COVID-19 vaccine (*B* = 1.54 OR = 4.65, *p* < 0.01). Those who held a bachelor’s degree or higher were also 157% more likely to receive the COVID-19 vaccine (*B* = 0.94, OR = 2.57, *p* = 0.02). Age and race were also significant covariates, with an increase in age leading to a 10% decrease in likelihood of COVID-19 vaccine acceptance (*B* = −0.11, OR = 0.90, *p* = 0.04). Those belonging to a minoritized population were 184% more likely than white participants to receive the vaccination (*B* = 1.04, OR = 2.84, *p* = 0.02). Additionally, participants who were pregnant at the time of the survey were 179% more likely to get vaccinated for COVID-19 compared to participants who were postpartum, though these results only approached significance at the 0.05 level (*B* = 1.03, OR = 2.79, *p* = 0.05).

## 4. Discussion

Within this study, 59% of participants had been vaccinated against COVID-19. This is much higher than the general pregnant population of the U.S. at the time, with a total at 31% [7]. When comparing group differences between participants who were and were not vaccinated for COVID-19, those with higher education were more likely to accept the COVID-19 vaccine. This finding is consistent with other vaccine acceptance studies [29], though contrary to what was seen in a COVID-19 vaccination study using the HBM in China [21]. Additionally, there were significant group differences in vaccine acceptance between those who had higher trust in their provider. This is consistent with research on vaccine acceptance during pandemic times that shows individuals who were more accepting of vaccines had higher trust in their health care system and their personal healthcare professionals [36]. Interestingly, knowing someone who had COVID-19, as opposed to having COVID-19 personally, was a significant group difference. While this finding is novel, others have noted that seeing someone be vaccinated leads to greater vaccine acceptance [37]. It is plausible that seeing someone sick, or even dying, from COVID-19 could be a strong motivator in accepting the vaccine to prevent serious illness and death. Interestingly, outside of education, trust in providers and knowing someone with COVID-19 were not statistically significant predictors for receiving the vaccination when controlling for other covariates in the model.

In our study, the HBM barriers and benefits constructs were indicative of COVID-19 vaccine acceptance. Perceived barriers were statistically significant, indicating that those who endorsed higher beliefs in the barriers were less likely to be vaccinated for COVID-19. This result is consistent with other research and our hypothesis [21]. Perceived benefits were statistically significant in predicting acceptance of the COVID-19 vaccines and was measured using the statement “I think that the shot for ‘COVID-19’ protects me.” This finding was consistent with our hypothesis. Other statistically significant covariates for vaccination were receiving all recommended vaccinations during pregnancy, higher education, being younger in age, and belonging to a minoritized population. Receiving all recommended vaccinations during pregnancy may indicate a trust in both the healthcare system and personal provider. This finding aligns well with other previous research on vaccine acceptance during pandemics [36]. In China, women of lower age were also more likely to accept the COVID-19 vaccine [21]. Additionally, reports and data show that most people in the U.S. who are unvaccinated are white [38]. Minoritized populations within the U.S. are vaccinated at levels similar to their total share of the population. Research has shown that throughout the pandemic, individuals of minoritized populations more rapidly came to believe that the vaccine helped protect them and keep them safe [39,40].

These finding indicate a need to improve health education with pregnant women regarding the benefits of the COVID-19 vaccine. Improving health education can include communicating the benefits of the vaccine to both the pregnant person and fetus at pre-natal care appointments; the use of motivational interviewing to create an environment in which the interviewer (a trained health professional) can discuss the patients questions, address concerns, and identify barriers in receiving the vaccine while pregnant; public health campaigns directed at pregnant women; increased science communication to the general public about the safety and benefits of the COVID-19 vaccine.

Barrier questions included: agreement that the vaccine was harmful toward the fetus, a COVID-19 vaccine has unpleasant side effects for the mother, reluctance to receive the COVID-19 shot, and reluctance toward receiving the flu shot. With the endorsement of these statements, it is likely the women are fearful of the side effects associated the vaccine for both herself and her fetus. Additionally, the vaccine is new and clinical trials have not included pregnant women, which may be a cause for concern for some of these women. It is important that healthcare providers provide patients with the most accurate information related to the safety and efficacy of the vaccine for pregnant women and their fetus. Healthcare providers should seek to understand why their patients are reluctant to receive the COVID-19 vaccine, discuss the patient’s concerns, and provide the patient with adequate information regarding the safety and efficacy of the COVID-19 vaccine. Pregnant people want to do what is best for their child, but in the age of instant information from both reliable and unreliable sources, it can be difficult for individuals without medical or research training to decipher what is good information or bad information. Through open and respectful communication, healthcare professionals can create an environment in which patients do not feel judgement for asking questions regarding the COVID-19 vaccines during pregnancy.

Finally, we recommend that the research community along with medical professionals continue to provide quality healthcare information to the general public in ways that are accessible to non-researchers. Publishing in open access journals, creating infographics, and short science communication videos are some of the ways in which we can do this. For academics, the authors opine the importance of being intentional in teaching students how to find reliable sources of information online, how to question sources, and how to seek out expert opinions.

Our study is not without limitations. First, the measure of HBM, while used in previous studies, is not a validated measure. The Cronbach’s alpha was moderate for these samples, and research on this measure and potential additional items used to capture the underlying construct of HBM should be expounded upon. Second, we obtained a convenience sample of women using social media as our recruitment outlet. While social media has become a popular recruitment technique [36,40], it limits who has access to the survey, often limiting those without social media profiles and those without internet access. However, social media is fairly popular and, therefore, it is possible we were able to capture a more diverse sample as a result. Additionally, we cannot confirm the participants received the vaccine; however, there was no incentive for participants to mislead the researchers about their vaccination status. The authors recognize that there may be other reasons for participants to receive the vaccine that are not captured within the HBM framework, such as societal pressures, job related mandates, and family, just to name a few. Finally, due to language limitations, the survey was made available only in English and thus limits those who are not fluent English speakers from participating in the study.

Despite the limitations, our findings are an important step to understanding COVID-19 vaccine acceptance among pregnant and postpartum women. Our sample included women in the U.S., and to our knowledge, no studies have examined COVID-19 vaccination acceptance using the HBM within this population. Second, our study was not connected to a health center, as are many studies that explore vaccine acceptance. This allowed a more general population to provide feedback. Those already connected with a health center may have greater trust in their provider or with the health system in general, potentially biasing results in favor of vaccinations [36]. Future studies should include multi-lingual surveys and expand recruitment efforts beyond social media.

## 5. Conclusions

The purpose of this study was to examine factors that relate to COVID-19 vaccine acceptance among pregnant and postpartum women in the United States. Using the HBM as our guiding framework, our study provides a timely assessment of COVID-19 vaccine acceptance among pregnant and postpartum women. Barriers related to receiving the vaccine were predictive of lower vaccine acceptance in this sample. Perceived benefits indicated higher acceptance of vaccination within this sample. Given the added harm caused to pregnant women by COVID-19, this research, along with other pertinent vaccine acceptance research, has the potential to inform future research goals and health interventions during the continuation of the COVID-19 pandemic and in future pandemics. Healthcare practitioners are encouraged to educate patients on the benefits of the COVID-19 vaccine.

## Figures and Tables

**Table 1 vaccines-10-00842-t001:** Questionnaire addressing Health Belief Model Dimensions.

Dimension	List of Statements
Perceived barriers	1. I think that the COVID-19 vaccine is harmful for my baby. 2. A COVID-19 vaccine will have unpleasant side-effects for me. 3. I do not want to get the COVID-19 vaccine. 4. I am usually against getting flu shots.
Perceived benefits	1. I think the vaccine for COVID-19 protects me.
Perceived susceptibility	1. I think I have a risk of acquiring COVID-19. 2. I think pregnant women would get sicker from COVID-19. 3. I think newborn children have a higher risk of getting COVID-19 than older children.
Perceived severity	1. I think pregnant women have a higher risk of getting COVID-19 than non-pregnant women. 2. I think the risk of COVID-19 is more severe to pregnant women than non-pregnant women. 3. COVID-19 is a serious disease. 4. COVID-19 is highly contagious. 5. I am concerned about the side-effects of COVID-19 for my baby.

**Table 2 vaccines-10-00842-t002:** Demographic characteristics of the sample by vaccination status.

Characteristic	Vaccinated (*n* = 134)	Not Vaccinated (*n* = 93)	Mann–Whitney U	*p* Value
Age in years	M = 29.96, SD = 4.19	M = 29.11, SD = 3.38	U = 6700.00	*p* = 0.201
Racial and Ethnic Identity, *n* (%) White Black/African American More than one Race Hispanic Asian Native American Alaska Native Other Prefer not to say	96 (71.6) 11 (8.2) 12 (9.0) 8 (6.0) 4 (3.0) 2 (1.5) 0 (0.0) 1 (0.7) 0 (0.0)	67 (72.0) 10 (10.8) 5 (5.4) 3 (3.2) 1 (1.1) 3 (3.2) 2 (2.2) 0 (0.0) 2 (2.2)	U = 5902.50	*p* = 0.350
Education, *n* (%) Some High School High School Diploma/GED Some College or Associates Bachelors Masters Doctoral	1 (0.7) 11 (8.2%) 37 (27.6%) 33 (24.6%) 25 (18.7%) 27 (20.1%)	2 (2.2%) 12 (12.9%) 46 (49.5%) 19 (20.4%) 11 (11.8%) 3 (3.2%)	U = 8334.50	*p* < 0.001
Pregnancy status, *n* (%) Pregnant Postpartum	108 (80.6) 26 (19.4)	78 (83.9) 15 (16.1)	U = 6027.00	*p* = 0.529
Number pregnancy, *n* (%) First Second Third Fourth Fifth More than fifth	69 (51.5) 46 (34.3) 11 (8.2) 2 (1.5) 4 (3.0) 2 (1.5)	53 (57.0) 28 (30.1) 8 (8.6) 3 (3.2) 1 (1.1) 0 (0.0)	U = 6582.00	*p* = 0.422
Past pregnancy loss, *n* (%) Yes No Unsure Prefer not to answer	34 (25.4) 98 (73.1) 1 (0.7) 1 (0.7)	19 (20.4) 73 (78.5) 0 (0.0) 1 (1.1)	U = 5946.50	*p =* 0.435
Prenatal provider, *n* (%) OBGYN Midwife Other Not answered	114 (85.1) 18 (13.4) 2 (1.5) 0 (0.0)	81 (87.1) 9 (9.7) 2 (2.2) 1 (1.1)	U = 6338.00	*p* = 0.546
Trust in provider to give accurate health information, *n* (%) Strongly Agree Agree Somewhat Agree Neither Agree nor Disagree Somewhat Disagree Disagree Strongly Disagree	61 (45.5) 44 (32.8) 21 (15.7) 6 (4.5) 1 (0.7) 1 (0.7) 0 (0.0)	25 (26.9) 33 (35.5) 21 (22.6) 10 (10.8) 2 (2.2) 1 (1.1) 1 (1.1)	U = 4712.50	*p* = 0.001
Received all recommended vaccines during pregnancy, *n* (%) Yes Most Some None	107 (79.9) 5 (3.7) 20 (14.9) 2 (1.5)	38 (40.9) 3 (3.2) 30 (32.3) 22 (23.7)	U = 3514.50	*p* < 0.001
Normally receive flu vaccine when not pregnant, *n* (%) Yes Most of the time Some of the time No Not answered	83 (61.9) 11 (8.2%) 25 (18.7%) 15 (11.2) 0 (0.0)	54 (58.1) 9 (9.7) 17 (18.3) 12 (12.9) 1 (1.1)	U = 5962.00	*p* = 0.633
Ever received HPV vaccine, *n* (%) Yes Maybe No Unsure	78 (58.8) 9 (6.7) 42 (31.3) 5 (3.7)	50 (53.8) 7 (7.5) 32 (34.4) 4 (4.3)	U = 5952.50	*p* = 0.518
Personal COVID-19 diagnosis, *n* (%) Yes No Not diagnosed but I think I had it	13 (9.7) 119 (88.8) 2 (1.5)	9 (9.7) 80 (86.0) 4 (4.3)	U = 6071.50	*p* = 0.560
Known someone personally who contracted COVID-19, *n* (%) Yes No	98 (73.1) 36 (26.9)	41 (44.1) 52 (55.9)	U = 4421.00	*p* < 0.001
Total Perceived Vaccine Barriers Total Perceived Vaccine Benefits Total Perceived Virus Severity Total Perceived Virus Susceptibility	M = 10.07, SD = 3.65 M = 4.26, SD = 0.93 M = 20.63, SD = 3.79 M = 11.16, SD = 2.30	M = 12.77, SD = 2.57 M = 3.34, SD = 0.97 M = 19.30, SD = 3.83 M = 10.64, SD = 2.37	U = 3737.50 U = 9441.00 U = 7659.50 U = 7179.00	*p* < 0.001 *p* < 0.001 *p* = 0.003 *p* = 0.049

**Table 3 vaccines-10-00842-t003:** Results of logistic regression predicting COVID-19 vaccine receipt with perceived benefits, barriers, susceptibility, and severity, controlling for pregnancy, education, receiving all recommended vaccines, knowing someone who contracted COVID-19, trust in medical providers, age, and race.

Variable	*B*	*SE*	*p*	*OR*	95% CI	Cohen’s *d*
(Intercept)	−0.08	2.079	0.97	0.92		
Perceived Benefits	0.66	0.23	<0.01	1.93	[1.23, 3.04]	0.36
Perceived Barriers	−0.20	0.07	0.01	0.82	[0.71. 0.95]	0.11
Perceived Susceptibility	0.03	0.11	0.83	1.03	[0.82, 1.28]	0.01
Perceived Severity	0.01	0.07	0.79	1.17	[0.37, 3.75]	0.01
Currently Pregnant	1.03	0.52	0.05	2.79	[1.01, 7.74]	0.57
Bachelor’s Degree or Above	0.94	0.39	0.02	2.57	[1.20, 5.48]	0.52
Received All Recommended Vaccines	1.54	0.40	<0.01	4.65	[2.12, 10.22]	0.84
Known Someone with COVID-19	0.36	0.40	0.37	1.44	[0.65, 3.17]	0.20
Trusts Medical Provider	0.16	0.59	0.79	1.17	[0.37, 3.75]	0.08
Age	−0.11	0.05	0.04	0.90	[0.81, 0.99]	0.06
Belonging to a Minoritized Race	1.04	0.44	0.02	2.84	[1.20, 6.73]	0.58

*
Notes.
*χ^2^ (11) = 88.58, *p* < 0.001, McFadden *R*^2^ = 0.29.

## Data Availability

The data from this study has been made publicly available at Harvard Dataverse: doi:10.7910/DVN/MHGEBV.

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
