# Peer review of "Using the Health Belief Model to Identify Predictors of COVID-19 Vaccine Acceptance among a Sample of Pregnant Women in the U.S.: A Cross-Sectional Survey"

_vaccines, 2022, doi:10.3390/vaccines10060842_

Round 1

Reviewer 1 Report

Although mass vaccination against COVID-19 may prove to be the most efficacious end to pandemic, there remain concern and indecision among the public toward vaccination. In this manuscript, the authors used the health belief model to identify factors that predict acceptance of the COVID-19 vaccine among pregnant women in US. And multiple logistic regression was used to evaluate the influence of measured factors on acceptance of vaccine. To further improve the manuscript, a minor comment was listed as below.

In table 2, the authors summarized many characteristics and separated by vaccination status. This allows authors to do more analysis in table 4. For example, the authors can compare different educational level, high school vs graduate; undergraduate school vs graduate. And the authors can also analysis age difference on acceptance of vaccine. Thus, the author might get more conclusions.

Author Response

Thank you for this feedback. Education was left as a binary between those with a college degree or higher vs. those without a college degree based on the directionality provided in the Mann-Whitney U tests. Adding additional categories may reduce power as there will be overlap between what is mutually exclusive and exhaustive. Age and race were added as additional control variables to increase power and both were found to be statistically significant. Results for these can be found in Table 2 and within the Results section. 

Reviewer 2 Report

Thank you for asking me to review this article. COVID-19 is a public health emergency of international concern. There is still no definitive cure for this highly transmittable illness. Immunization and breaking the chain of infection is the only successful approach to mitigate its spread. Therefore, understanding the determinants of vaccination hesitancy is a useful tool for implementing strategic measures aimed at improving patient compliance with vaccination with particular reference to anti-COVID-19 vaccination. In this context, the paper under review is aimed at investigating factors that predict acceptance of the COVID-19 vaccine among pregnant women in US.

The subject under study is certainly very important, especially in the historical period we are experiencing. The article presents interesting results but, but it must be improved before publication, especially for its local impact, very small sample and different epidemiological context (June 2021).I would like to encourage authors to consider several issues to be improved.

Title: it must be improved, highlithening the main object, precise place and that the study is using a sample.

Introduction: The authors should make clearer what is the gap in the literature that is filled with this study? The authors do not frame their study within the vast body of literature that addressed the issue of the international situation regarding the acceptance of the vaccination in the different subgroups of adult population.

Methods: The survey was conducted using a non-standard questionnaire. The use of an unreliable instrument is a serious and irreversible limitation of the study. The fact that a similar questionnaire has been used in previous surveys is not sufficient, also because the original questionnaire was modified. A validation process must be performed to evaluate the tool in a different population. What about face validity, reliability and intelligibility? Was a preliminary pilot study conducted?

The enrolment procedure must be better specified. How did the authors choose the way to select the sample? This can represent a great bias origin. How did they avoid the selection bias? The author proposed a minimum sample size (114 subjects), but it is not clear what is the reference population? How large is it? Without the numerical identification of the reference population is not clear the meaning of this sample size.

Ethical Issue: the author stated that “The University of South Dakota (AJ’s previous institution) IRB approved the research”, but it is not clear if this institution is an officially authorized ethical competent body. Please declare and add approval number, since an ethical approval is necessary.

Statistical analysis: I suggest to insert a measure of the magnitude of the effect for the comparisons. Please consider to include effect sizes.

Discussion: I also suggest expanding. What is the possible international contribution of the study to the literature? What are the implications of the study? Emphasize the contribution of the study to the literature. The discussion must be updated with one of the principal debated argument in this epidemiological context: the use of a green pass linked to vaccination practice (refer to articles with DOI: https://doi.org/10.3390/vaccines9111222) and studies conducted in the same epidemiological period.

Author Response

Thank you for your feedback. Our responses are attached. 

Reviewer 3 Report

1) It might be better to include: A Cross-Sectional Survey in the title

2) Introduction should be longer. I suggest shortening the COVID-19 situation; I would elaborate more on the reasons of using the HBM for pregnant women.

3) It seems that the IRB approval number is missing.

4) Full survey Qs should be uploaded as a supplement.

5) The sub-section of the Sample size calculation is needed; in addition, the utilization of G-power needs to include more details.

6) Recruitment processes of Facebook and Twitter needs to be more detailed-oriented. Captures of the recruitment posts are valuable. Were there any awards for the participants? 

7) Would it be feasible to mention any strengths in the Discussion section? 

8) Conclusion section is too short -- it needs to be well-summarized section. 

9) Check the reference numbering again within the MN. Also, it seems that numbering in the [ ] is the correct way.

Author Response

Thank you for your feedback. We have attached our responses in a word document. 

Round 2

Reviewer 2 Report

The paper was improved

Reviewer 3 Report

Thank you for making corrections. I would suggest reviewing the entire MN again and adjust grammatical errors.